# Bloom Time Effect Depends on Muscle Type and May Determine the Results of pH and Color Instrumental Evaluation

**DOI:** 10.3390/ani11051282

**Published:** 2021-04-29

**Authors:** Damian Knecht, Kamil Duziński, Anna Jankowska-Mąkosa

**Affiliations:** Institute of Animal Breeding, Faculty of Biology and Animal Sciences, Wroclaw University of Environmental and Life Sciences, 51-630 Wroclaw, Poland; damian.knecht@upwr.edu.pl (D.K.); kamilduzinski@wp.pl (K.D.)

**Keywords:** pig carcass, muscles, bloom time, pH, color

## Abstract

**Simple Summary:**

The color of fresh pork is a very important feature, both for the meat industry and especially for direct consumers. Consumers often relate color to the freshness and quality of pork. Quality evaluation of fresh pork intended for sale or processing seems to be a priority task for meat plants due to the expectations of modern consumers. Meat can be characterized as a mixture of different chemical components, each of which contributes to its quality either independently or in combination with other ingredients. The complexity of muscle structure and the tasks performed by individual muscles may contribute to uneven color distribution. The presented results are an important contribution to the rapid and precise instrumental evaluation of pH and color. The information may be particularly important for meat plants, which, by introducing such an evaluation, allow consumers to additionally verify the product.

**Abstract:**

The aim of this study was to determine the effect of 30 min bloom time and the type of muscle on pH and color parameters together with the possibility of estimating these measurements. The research material consisted of 270 samples from 6 muscle types: LD—*Longissimus* *dorsi*, LL—*Longissimus lumborum*, IL—*Iliacus*, SEM—*Semimembranosus*, CT—*Cutaneous trunci*, LTD—*Latissimus dorsi*. Measurements included pH and color of fresh pork at 0 min, and after 30 min bloom time. Bloom time influenced all analyzed parameters, although to a varying effect, depending on the muscle type. The lowest pH values were noted for dorsal-located muscles (LD, LL), then in the ham area (IL, SEM), and the highest values of the location on the side surface of the carcass (CT, LTD). The large increase in the proportion of L* and a* was observed for CT muscle (20–30%, the highest of all observed) and LTD (20–25%); for LD and LL the largest growth changes were observed for parameters b* (15–20%) and H* (20–30%). The lowest number of strong correlations was noted for LD and CT muscles, and the largest for SEM. A very good fit (R^2^ > 0.90) of regression equations was achieved in 7 cases. The presented results are an important contribution to the rapid and precise instrumental evaluation of pH and color.

## 1. Introduction

The color of fresh pork is a very important feature, both for the meat industry and especially for direct consumers. Consumers often relate color to the freshness and quality of pork [1].

However, the evaluation of color by consumers is usually based on visual evaluation with the naked eye; while for research, scientific or industrial needs, a precise instrumental evaluation is much more significant. It is worth remembering that the evaluation of quality by consumers has been proven to be not objective, burdened with an extremely large number of external and internal factors shaping the personal perception [2,3,4].

The complexity of muscle structure and the tasks performed by individual muscles may contribute to uneven color distribution [5]. Pork half-carcass consists of approximately 100 striated muscles, and five of them make up almost a third of the musculature weight of the carcass [6]. LD—*Longissimus dorsi*, LL—*Longissimus lumborum* and SEM—*Semimembranosus* muscles have been evaluated most often, whereas muscles from the abdominal location have been subjected to evaluation much less frequently. However, due to the increasing commercial value of the belly, this is an element that should be taken into particular account [7]. The meat industry is interested in all the muscles but especially those included in the most valuable cuts [8]. Furthermore, anatomical and spatial imaging of changes gives an illustrative picture of the whole carcass.

It is considered that after 24 h of cooling down, changes in color are completed [9], although bloom time is essential for precise measurement methodologies [10]. Bloom time can have a major impact on fresh instrumental color readings because the oxidation of myoglobin pigment in the muscle is responsible for creating a red color and does not occur immediately [11]. Blooming takes place on muscle’s cut surface during the action of the air and allows the reaction of reduced purple myoglobin and hemoglobin on the muscle surface with oxygen to transform it into bright red oxygenated forms [12,13]. Inactivation of oxygen-absorbing enzymes gradually reduces the rate of change in color [14]. When the meat pigment metmyoglobin (MetMb) is oxidized, it causes a grayish-brown color to appear on the surface of the meat. In such situations, consumers judge the meat as stale and do not purchase it [15].

Research indicates that bloom time for meat should range from 20 min to 1–2 h [10,16,17], although in the case of pork, 30 min is deemed to be sufficient [12]. However, studies on the blooming time effect on pork were initially rare; most research activity in this area was conducted in 2001–2008. Even so, reviewing the research results brought ambiguous conclusions [5]. It seems that constant changes in the population of slaughtered pigs with regard to genetic and environmental factors causes the need to conduct ongoing research on this subject. Hence, the need to accurately evaluate the variability of pH and muscle color under industrial conditions.

The aim of this study was to determine the effect of 30 min bloom time and the type of muscle on pH and color parameters as well as to explore the possibility of predicting this effect.

## 2. Materials and Methods

### 2.1. Design of the Study

The research material consisted of: LD—*Longissimus dorsi*, LL—*Longissimus lumborum*, IL—*Iliacus*, SEM—*Semimembranosus*, CT—*Cutaneous trunci* and LTD—*Latissimus dorsi*, obtained in the same way from the same pig carcasses.

The animals were transported in special livestock transport vehicles over a distance of max. 35 km. The animals were slaughtered at the age of 6.5–7 months and at a body weight of approx. 110 kg in a meat plant in Wielkopolskie province, using the electric stunning method. Pre-slaughter and slaughter time were similar for all the animals (the time of rest and fasting was about 4 h at a temperature of about 17 °C). The carcasses were bled, separated along the center line and deprived of tongue, bristle, hooves, genital organs, perirenal fat, kidneys, diaphragm, eyes, middle ear, brain and spinal cord.

The muscles intended for further analysis were cut from right carcasses after 24 h of cooling, according to the production methodology in force at the slaughterhouse. The bloom time samples were collected immediately after dissection of the muscle in question, cut perpendicularly to the width of the muscle. The samples for testing were vacuum-packed and transported in stable cooling conditions directly to the meat evaluation laboratory at the Institute of Animal Science, Wroclaw University of Environmental and Life Sciences, where they were then subjected to further analysis.

The cooling of carcasses was performed first by holding them for 2 h at −15 to −8 °C, followed by storage at 0–2 °C (8–12 h) until carcass temperature dropped to 5 °C. After 24 h of cooling, the abovementioned muscles were cut from the same right half-carcasses and marked. In total, 270 samples were obtained.

### 2.2. Meat Quality Assessment

Bloom time was determined using measurements at 0 and 30 min. The 0 min measurement was classified as the measurement of fresh meat; 30 min bloom time took place in dark and cool conditions (4 °C).

pH measurements were conducted for each muscle location immediately after cross-division of the element with a Testo 205 pH meter with gel electrolyte (manufactured by Testo Sp. z o. o., Warsaw, Poland). Instrumental color evaluations were performed with a Minolta Chroma CR 400 device (Konica, Osaka, Japan) with an 11 mm diameter aperture, D65 illuminant, calibrated against a white tile. The measurements were made three times for each muscle. During instrumental color evaluation, the following parameters were determined: CIE—Commission Internationale de l’Eclairage L*, a* and b*—values representing lightness, redness and yellowness, respectively. Additionally, Chroma (C*) and Hue angle (H*) indicators were determined. Chroma described the color saturation in the CIE L*a*b* space and was calculated according to the formula C* = (a*^2^ + b*^2^)^1/2^. Hue angle determined the immaculateness of color and was calculated as H* = (tan^−1^ b*/a*) [18]. The percentage change in color over the 30 min bloom time period was determined using the formulas developed by [14].

### 2.3. Statistical Analysis

The data analysis was based on factor analysis in which two main effects of muscle and bloom time were observed. Additionally, differences between muscle samples at 0 and 30 min bloom time were separately compared. Correlation coefficients between qualitative parameters within each location were calculated using r-Pearson correlation. Models of regression equations estimating particular pH and color parameters at 30 min bloom time compared to measurements at 0 min were based on correlations with the dependent variable by the stepwise method to obtain the lowest estimation error. The accuracy of the estimation was determined by means of prediction error and variable matching. F, R^2^ determination coefficient, *p*-value and estimation error were used to describe the equations.

## 3. Results

The changes in pH depending on the bloom time (0 and 30 min) are presented in Figure 1. All pH_0min_ values were observed on the level of correct values for RFN pork (meat of normal quality—reddish-pink, firm, normal, non-exudative) after 24 h from slaughter. However, differences between the analyzed individual muscles were noted. Furthermore, the differences were also partially confirmed statistically, irrespective of the pH value measurement at 0 or 30 min. The lowest pH values were noted for dorsal-located muscles (LD, LL) and those located in the ham area (IL, SEM), while the highest pH values were for the location on the side surface of the carcass (CT, LTD). No statistically significant differences were found in the anatomically similar locations for pH_30min_, but for pH_0min_, differences were reported in ham and side locations.

An increase in all pH values after 30 min bloom time was observed regardless of the muscle type, although the lowest increase was found for LD and LL below 0.11 (*p* ≤ 0.05), and the highest for IL and SEM over 0.20 values (*p* ≤ 0.01).

Color changes with respect to bloom time and muscle type are presented in Table 1. The distribution of differences between muscles was not uniform, both within individual color parameters and for the two bloom time periods. Differences at a statistically confirmed level (*p* ≤ 0.05) after 30 min bloom time were observed only for LD muscle (all color parameters), LL (b*, C*, H*) and SEM (L*, a*, C*). Color measurements at 0 min differed (*p* ≤ 0.01) between muscles for L*, a*, C*, H*. Color measurements at 30 min differed (*p* ≤ 0.01) between muscles for L*, a*, b*, H*. Regardless of bloom time, always two LL and SEM muscles alternately obtained the highest and lowest values of the evaluated color parameters.

The percentage share of color changes during 30 min bloom time is shown in Figure 2. A percentage increase of color parameters was noted for almost all the muscles (except SEM and partially IL), although to a varying degree. These changes were not homogeneous and were subject to individual variability. However, it should be noted that for LD and LL, the largest growth of changes was observed for parameters b* and H* (between 20–30% for the former muscle and 15–20% for the latter). The percentage decrease in the SEM muscle color value was at a level not exceeding 5%, with the dominant a* and C* parameters. As it was the case for LD and LL muscles, the highest variability of color in the IL muscle was observed for b* and H*, although it exceeded 5% only slightly. A large increase in the proportion of L* and a* was observed for CT muscle (20–30%, the highest of all observed) and LTD (20–25%).

Figure 3 presents the dependence of percentage color change on pH parameters after 30 min bloom time without the division for muscles. A general tendency emerging from individual observations does not indicate any unambiguous dependence in this respect. Therefore, there seems to be no dependency between bloom time effect and the changing muscle pH, nor the interaction of these two main effects. The greatest dispersion was recorded for the b* and H* parameters. On the other hand, the highest stability was observed for the L* and a* parameters.

The correlation coefficients for pH and color parameters for the individual muscle types at 0 min and 30 min bloom time are presented in Table 2. Numerous correlation coefficients were observed at moderate and strong levels, which were statistically confirmed (*p* ≤ 0.05, *p* ≤ 0.01). The lowest number of such correlations was noted for LD and CT muscles, and the highest for SEM.

Regression equations estimating the pH values and color parameters after 30 min bloom time on the basis of measurements at 0 min were obtained by progressive regression method (Table 3). The accuracy of the obtained equations varied depending on the estimated parameter and muscle. An observation that connected all the equations was the use of more than one constituent parameter (except equations a* and b* for LD). A very good fit (R^2^ > 0.90) of regression equations was achieved in 7 cases (2 for LD—pH and L*; 1 for LL—H*; 1 for IL—pH; 1 for SEM—pH; 1 for LTD—pH; 1 for overall—pH). The adjustment at an unsatisfactory level (R^2^ < 0.50) was noted in 6 cases (1 for SEM—a*; 4 for LTD—a*, b*, C*, H*; 1 for overall—C*). Within the individual muscles, the equations for LTD were burdened with the biggest error of estimation, whereas those for LL were most precise.

## 4. Discussion

We found that the affinity of color variability due to bloom time is dependent on the type of muscle and location (especially in the case of dorsal and ventral muscles). The measurements were taken immediately after the cutting of the carcasses into material that was sold directly, in an unprocessed form, which undoubtedly influenced the practical aspect of the experiment. Our previous studies indicated that differences in carcass quality parameters may have a pronounced effect on the efficiency of meat production [19]. Meat plants are interested in the evaluation of the color of meat as a whole commercial cut [20]. Meat color is one of the main quality parameters, and as a sensory feature, affects the price of freshly cut parts and the final quality of processed pork products [21].

SEM and IL muscles, located in the ham part, were characterized by the highest stability of color during 30 min bloom time. Other muscle types were characterized by similarly high variability, although it concerned various parameters. LD and LL muscles were found to change (increase) the proportion of the yellow color and H* the most, whereas CT and LTD muscles—the proportion of light and red color. High surface oxygen consumption by tissue is associated with poor oxygen penetration [22], which explains the clear disproportions of results for different muscle types. Since high variability of oxidative enzyme activity was observed for beef muscles [23], it may be presumed that similar dependences exist in the case of pork meat.

The color of meat is determined by many factors, including the pH level that affects protein structure and moisture [24,25]. The connection between pH changes and stabilization and color variability was noted in earlier studies [26,27]. In our study, comparing the variability of color parameters with pH changes, the lowest average pH value was obtained for muscles located in the ham area. This is the opposite of what was reported by Zhu and Brewer [28], who pointed to the highest color variation for muscles with low pH, not those with high pH. However, it should be noted that our research was carried out on a research population with no quality defects. On the other hand, additional observations did indicate dependences in the change in pH over a 30 min period: SEM and IL muscles, in which the increase of pH was the highest, were characterized by the smallest variability of color (less than 5%) regardless of the tested parameters. Similarly, the muscles with the lowest increase of pH were characterized by the highest variability, although this concerned different color parameters.

The distribution of individual observations indicates that pH determines the color although it depends on the specific color parameter. Most often, the best fit for regression equations was characterized by the equations developed for pH_30min_, always using the pH_0min_. The analysis of the developed regression equations indicates that the equations developed for the LTD muscle were most inaccurate. This emphasizes the need to pay special attention to changes occurring in this muscle during bloom time, because these changes cannot be easily predicted. Our previous studies clearly indicate that muscles from the belly cut are most difficult to evaluate qualitatively [29,30].

The saturation index (C*), which refers to the intensity of color, in most cases was the only color parameter that did not significantly correlate with the L* parameter, which measures the lightness of the color compared to the white standard. Some authors indicate that this parameter shows the least variability during bloom time [10,12]. It is partially consistent with our results, but with the exception of CT and LTD muscles (again located abdominally). Škrlep and Čandek-Potokar [12] reported that the highest variability during bloom time was observed for b* parameter. We confirm this observation for dorsal location of muscles (LD and LL) and even IL muscle.

Additionally, in the case of the loin muscle, the correlations demonstrated in this study were different from previous findings [12], and definitely with a different intensity. We showed a surprisingly large number of weak correlations for the tested pH parameters and color before and after bloom time for the LD muscle, which, after all, is treated as a model muscle for quality analysis. However, recent research clearly indicates that using the quality parameters of loin for the entire carcass is doubtful [30,31].

As is the case with beef, we agree with the statement of Holman et al. [32], that there are still no objective instrumental criteria that could be used by consumers to evaluate pork color. The color difference in the case of meat with quality defects was the basis for a deeper visual analysis of fresh elements, hence the development of research in this area [10]. However, the distribution of meat quality defects in the pig carcass is not homogeneous for all skeletal muscles [33]. Our study focused exclusively on muscles without qualitative defects, which allowed us to visualize changes caused by 30 min bloom time. In a wider context, it should be considered that even muscles from the same carcass behave differently.

## 5. Conclusions

Although bloom time influenced all the analyzed parameters, its effect was different depending on the muscle type. The presented results constitute an important contribution to the quick and precise instrumental evaluation of pH and color. The information may be of particular importance for slaughterhouses which, by introducing this assessment, could allow consumers to additionally verify the product. The contemporary consumer pays ever greater attention to the quality of consumed products and subjects, and evaluates them increasingly carefully. Only products characterized by appropriate quality and health safety are chosen. The quality of pig meat has long been of interest to both scientists and food technologists. We found that the affinity of color variation with bloom time was dependent on the type of muscle and its location (especially dorsal and abdominal muscles). SEM and IL muscles, located in the ham part, were characterized by the highest stability of color during 30 min bloom time. Other muscle types were characterized by similarly high variability, although it concerned various parameters. The quality of the finished product and the profitability of a meat processing plant depends, to a large extent, on the quality of raw materials. The nature of color variation can also be predicted from the regression equations proposed by us, which constitute a convenient computational tool. To recapitulate, we presented additional possibilities of quick and precise sorting of raw materials intended for consumer sale.

## Figures and Tables

**Figure 1 animals-11-01282-f001:**
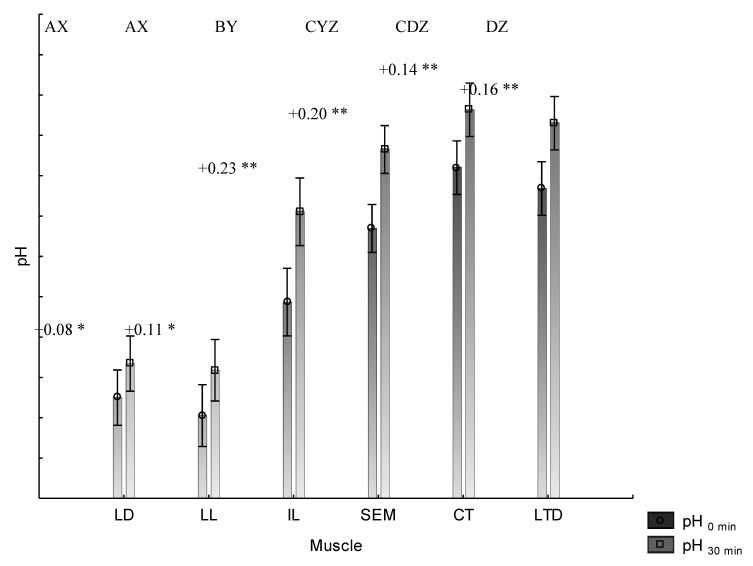
Changes in pH measurements including bloom time and muscles. * *p* ≤ 0.05, ** *p* ≤ 0.01—difference between pH_0min_ and pH_30min_ within the same muscle; ^A,B,C,D^—differences between muscles for pH_0min_, *p* ≤ 0.01; ^X,Y,Z^—differences between muscles for pH_30min_, *p* ≤ 0.01; LD—*Longissimus dorsi*, LL—*Longissimus lumborum*, IL—*Iliacus*, SEM—*Semimembranosus*, CT—*Cutaneous trunci*, LTD—*Latissimus dorsi*.

**Figure 2 animals-11-01282-f002:**
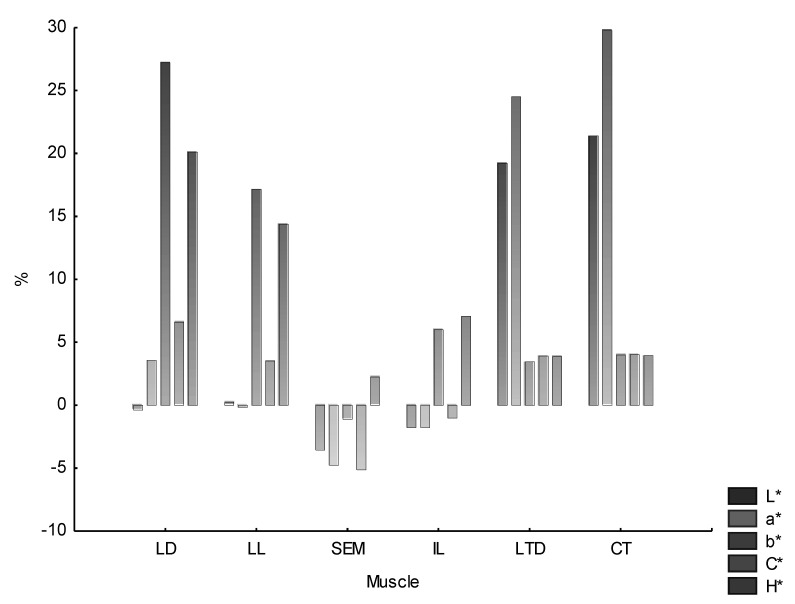
The percentage increase/decrease of color parameters during bloom time depending on the muscle type. LD—*Longissimus dorsi*, LL—*Longissimus lumborum*, IL—*Iliacus*, SEM—*Semimembranosus*, CT—*Cutaneous trunci*, LTD—*Latissimus dorsi*.

**Figure 3 animals-11-01282-f003:**
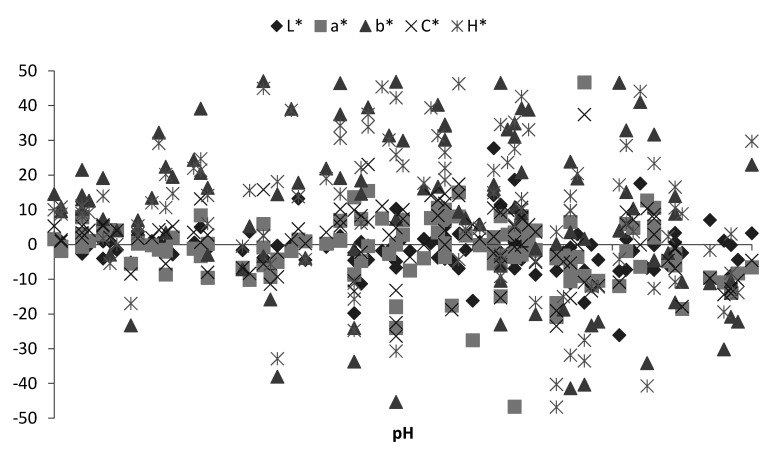
Dependences of percentage color change and pH during 30 min bloom time.

**Table 1 animals-11-01282-t001:** Changes in color parameters with regard to bloom time and muscle type.

Item	Color				
	L* Mean ± SE	a* Mean ± SE	b* Mean ± SE	C* Mean ± SE	H* Mean ± SE
LD					
0 min	55.52 ^B^ ± 4.75	15.12 ^AB^ ± 1.01	6.43 ± 1.54	16.49 ^A^ ± 1.11	22.95 ^AB^ ± 5.05
30 min	55.29 ^Y^ ± 4.66	15.65 ^XY^ ± 1.11	7.89 ^YZ^ ± 1.01	17.55 ± 1.09	26.79 ^YZ^ ± 3.25
LL					
0 min	59.55 ^A^ ± 3.91	14.63 ^A^ ± 1.13	7.74 ^B^ ± 1.30	16.59 ^A^ ± 1.19	27.86 ^B^ ± 4.19
30 min	59.67 ^X^ ± 3.69	14.64 ^X^ ± 1.52	8.99 ^Z^ ± 1.21	17.21 ± 1.67	31.61 ^Z^ ± 3.24
IL					
0 min	50.67 ^C^ ± 2.71	16.91 ^BCD^ ± 1.23	6.39 ± 1.43	18.11 ^ABC^ ± 1.41	20.64 ^A^ ± 4.03
30 min	49.81 ^Z^ ± 4.21	16.57 ^YZ^ ± 1.07	6.65 ^XY^ ± 1.54	17.89 ± 1.39	21.68 ^XY^ ± 4.08
SEM					
0 min	46.46 ^D^ ± 4.44	18.19 ^D^ ± 1.54	6.87 ± 2.72	19.55 ^C^ ± 2.35	20.09 ^A^ ± 5.96
30 min	44.67 ^V^ ± 3.49	17.27 ^Z^ ± 1.50	6.25 ^X^ ± 2.35	18.44 ± 2.18	19.41 ^X^ ± 5.21
CT					
0 min	51.61 ^C^ ± 5.69	17.22 ^CD^ ± 3.76	6.66 ± 2.81	18.71 ^BC^ ± 3.62	21.38 ^A^ ± 10.15
30 min	51.39 ^Z^ ± 6.91	16.46 ^YZ^ ± 3.21	7.02 ^XY^ ± 3.04	18.13 ± 3.31	22.77 ^XY^ ± 9.91
LTD					
0 min	50.71 ^C^ ± 4.19	16.74 ^C^ ± 2.79	5.88 ^A^ ± 2.82	17.92 ^AB^ ± 3.04	18.83 ^A^ ± 8.42
30 min	49.51 ^Z^ ± 5.11	16.09 ^Y^ ± 2.16	6.39 ^XY^ ± 3.33	17.64 ± 2.09	21.41 ^X^ ± 11.37

^A,B,C,D^—differences between muscles (the same column) for color values with bloom time 0 min, *p* ≤ 0.01. ^X,Y,Z,V^—differences between muscles (the same column) for color values with bloom time 30 min, *p* ≤ 0.01; LD—*Longissimus* *dorsi*, LL—*Longissimus lumborum*, IL—*Iliacus*, SEM—*Semimembranosus*, CT—*Cutaneous trunci*, LTD—*Latissimus dorsi*, means ± SE, L*— white, a*—red, b*—yellow, C*—scale values CIE, H*—scale values CIE.

**Table 2 animals-11-01282-t002:** Correlation coefficients between pH and color parameters at 0 min and 30 min bloom time for each muscle type.

Item		pH_0min_	pH_30min_	L*	a*	b*	C*	H*	L*_30min_	a*_30min_	b*_30min_	C*_30min_	H*_30min_
LD/	pH_0min_	-	0.94 **	−0.18	−0.21	−0.68 **	−0.55 *	−0.60 *	−0.11	−0.19	−0.55 *	−0.39 **	−0.39 *
LL	pH_30min_	0.93 **	-	−0.04	−0.14	−0.64 **	−0.45 *	−0.58 *	0.02	−0.07	−0.53 *	−0.27	−0.44 *
	L*	−0.18	−0.13	-	−0.20	0.27	−0.02	0.30 *	0.98 **	−0.10	0.21	−0.01	0.24
	a*	0.06	0.13	0.25	-	−0.04	0.83 **	−0.30 *	−0.16	0.81 **	−0.12	0.69 *	−0.50 *
	b*	−0.56 *	−0.53 *	0.15	−0.03	-	0.51 *	0.97 **	0.24	−0.08	0.73 **	0.22	0.68 **
	C*	−0.24	−0.16	0.28	0.85 **	0.50 *	-	0.28	−0.01	0.65 **	0.29 *	0.71 *	−0.06
	H*	−0.55 *	−0.56 *	0.04	−0.45 *	0.91 **	0.09	-	0.26	−0.28	0.72 **	0.03	0.77 **
	L*_30min_	−0.21	−0.15	0.92 **	0.27	0.21	0.33 *	0.08	-	−0.12	0.20	−0.03	0.23
	a*_30min_	0.07	0.14	0.25	0.90 **	0.05	0.81 **	−0.34 *	0.30 *	-	0.01	0.92 *	−0.47 *
	b*_30min_	−0.35 *	−0.30 *	0.30 *	0.22	0.85 **	0.63 **	0.66 **	0.38 *	0.39 *	-	0.41 **	0.88 **
	C*_30min_	−0.08	0.00	0.31 *	0.80 **	0.36 *	0.88 **	−0.02	0.38 *	0.94 **	0.69 **	-	−0.08
	H*_30min_	−0.42 *	−0.43 *	0.11	−0.50 *	0.80 **	−0.01	0.92 **	0.15	−0.39 *	0.69 **	−0.05	-
IL/	pH_0min_	-	0.94 **	−0.30	−0.36	−0.58 *	−0.49 *	−0.51 *	−0.41 *	−0.52 *	−0.71 **	−0.65 **	−0.68 **
SEM	pH_30min_	0.96 **	-	−0.29	−0.31	−0.62 **	−0.47 *	−0.56 *	−0.40 *	−0.45 *	−0.70 **	−0.60 *	−0.68 **
	L*	−0.24	−0.19	-	−0.27	0.44 *	−0.07	0.52 *	0.80 **	−0.30	0.33	−0.07	0.41 *
	a*	−0.48 *	−0.47 *	0.30 *	-	0.34	0.95 **	0.01	0.05	0.74 **	0.29	0.65 **	0.11
	b*	−0.34 *	−0.33 *	0.63 **	0.82 **	-	0.62 **	0.94 **	0.51 *	0.37 *	0.61 **	0.51 *	0.59 *
	C*	−0.45 *	−0.43 *	0.47 *	0.97 **	0.94 **	-	0.32	0.22	0.74 **	0.44 *	0.71 **	0.29
	H*	−0.28	−0.26	0.64 **	0.74 **	0.98 **	0.87 **	-	0.49 *	0.16	0.57 *	0.34	0.61 **
	L*_30min_	−0.35 *	−0.31 *	0.79 **	0.39 *	0.63 **	0.52 *	0.61 **	-	0.13	0.61 **	0.36	0.60 *
	a*_30min_	−0.56 *	−0.55 *	0.26	0.55 *	0.39 *	0.51 *	0.31 *	0.35 *	-	0.55 *	0.94 **	0.33
	b*_30min_	−0.64 **	−0.63 **	0.41 *	0.63 **	0.62 **	0.67 **	0.55 *	0.59 *	0.81 **	-	0.81 **	0.97 **
	C*_30min_	−0.62 **	−0.60 *	0.36 *	0.61 **	0.52 *	0.61 **	0.44 *	0.49 *	0.97 **	0.93 **	-	0.63 **
	H*_30min_	−0.66 **	−0.65 **	0.35 *	0.62 **	0.62 **	0.67 **	0.56 *	0.56 *	0.69 **	0.98 **	0.84 **	-
CT/	pH_0min_	-	0.92 **	−0.02	−0.07	0.04	−0.05	0.08	−0.06	0.06	0.18	0.10	0.19
LTD	pH_30min_	0.96 **	-	−0.02	−0.11	−0.01	−0.11	0.05	−0.08	0.05	0.17	0.10	0.20
	L*	0.08	0.15	-	−0.42 *	0.67 **	−0.21	0.84 **	0.86 **	−0.31 *	0.69 **	0.00	0.83 **
	a*	−0.06	−0.13	−0.30 *	-	0.18	0.97 **	−0.36 *	−0.20	0.72 **	0.08	0.62 **	−0.29 *
	b*	0.07	0.03	0.45 *	0.34 *	-	0.42 *	0.84 **	0.74 **	0.03	0.82 **	0.31 *	0.77 **
	C*	−0.04	−0.10	−0.10	0.95 **	0.60 *	-	−0.12	0.02	0.67 **	0.29 *	0.65 **	−0.05
	H*	0.14	0.11	0.53 *	0.08	0.96 **	0.37 *	-	0.78 **	−0.35 *	0.73 **	−0.03	0.88 **
	L*_30min_	−0.10	−0.09	0.36 *	0.09	0.64 **	0.29 *	0.68 **	-	−0.36 *	0.66 **	−0.06	0.78 **
	a*_30min_	0.29 *	0.23	−0.25	0.60 **	0.18	0.56 *	0.01	−0.01	-	0.25	0.93 **	−0.17
	b*_30min_	0.03	0.02	0.40 *	−0.12	0.57 *	0.08	0.62 **	0.60 *	−0.11	-	0.58 *	0.90 **
	C*_30min_	0.25	0.19	−0.01	0.47 *	0.47 *	0.55 *	0.34 *	0.30 *	0.80 **	0.50 *	-	0.19
	H*_30min_	−0.02	−0.01	0.42 *	−0.26	0.44 *	−0.07	0.54 *	0.53 *	−0.39 *	0.95 **	0.23	-

*—correlation statistically significant with *p* ≤ 0.05, **—correlation statistically significant with *p* ≤ 0.01. LD—*Longissimus* *dorsi*, LL—*Longissimus lumborum*, IL—*Iliacus*, SEM—*Semimembranosus*, CT—*Cutaneous trunci*, LTD—*Latissimus dorsi*, L*— white, a*—red, b*—yellow, C*—scale values CIE, H*—scale values CIE.

**Table 3 animals-11-01282-t003:** Regression equations estimating pH and color parameters after 30 min bloom time on the basis of fresh samples and evaluation for each muscle type.

Item		pH	L*	a*	b*	C*	H*	Intercept	R^2^	Estimation Error	F
LD	pH_30min_	0.86	0.01	0.02	-	-	-	0.11	0.91	0.06	131.58
	L*_30min_	2.26	0.98	-	-	0.32	-	−16.31	0.97	0.88	399.55
	a*_30min_	-	-	0.89	-	-	-	2.18	0.65	0.66	79.95
	b*_30min_	-	-	-	0.48	-	-	4.83	0.54	0.69	49.83
	C*_30min_	−0.84	-	-	-	0.69	−0.06	11.92	0.56	0.76	17.29
	H*_30min_	-	−0.07	-	12.71	−5.61	−2.89	107.62	0.74	1.72	29.21
LL	pH_30min_	0.82	-	0.01	-	-	-	0.89	0.87	0.07	114.17
	L*_30min_	-	0.86	-	0.22	-	-	6.58	0.86	1.42	101.19
	a*_30min_	-	-	2.73	4.84	−3.47	−1.07	24.68	0.84	0.64	41.68
	b*_30min_	0.63	-	5.23	8.08	−7.57	−1.24	26.93	0.87	0.47	39.63
	C*_30min_	1.22	0.04	-	-	1.26	-	−12.12	0.80	0.77	43.63
	H*_30min_	1.26	0.08	5.83	10.43	−10.07	−1.23	55.84	0.90	1.13	43.13
IL	pH_30min_	0.95	-	-	-	-	−0.01	0.66	0.90	0.11	115.65
	L*_30min_	-	1.36	0.99	-	-	-	−36.37	0.72	2.29	35.18
	a*_30min_	−0.88	−0.05	1.42	−2.28	-	0.79	−1.63	0.72	0.62	12.55
	b*_30min_	−2.83	-	-	0.32	-	-	20.18	0.56	1.06	17.31
	C*_30min_	−2.25	−0.09	-	-	0.45	-	26.69	0.65	0.87	15.79
	H*_30min_	−6.86	-	-	-	-	0.37	51.72	0.56	2.81	16.97
SEM	pH_30min_	0.99	0.01	-	-	-	-	0.12	0.93	0.08	371.16
	L*_30min_	−1.29	0.55	-	-	0.22	-	22.22	0.67	2.05	38.03
	a*_30min_	−1.93	-	0.36	-	-	-	21.69	0.42	1.16	20.69
	b*_30min_	−3.23	-	−1.68	-	1.89	−0.16	21.84	0.63	1.48	23.36
	C*_30min_	−2.58	0.10	-	-	0.67	−0.15	18.31	0.55	1.51	16.83
	H*_30min_	−8.31	−0.16	−5.79	−1.26	6.18	-	67.31	0.65	3.23	19.95
CT	pH_30min_	1.04	-	-	-	−0.01	-	−0.02	0.86	0.07	133.65
	L*_30min_	-	0.79	-	0.74	-	-	5.42	0.78	3.29	81.26
	a*_30min_	2.99	0.22	2.12	-	−1.47	-	−21.53	0.57	2.19	14.35
	b*_30min_	3.19	0.16	−3.29	−0.59	3.79	-	−30.15	0.79	1.48	31.05
	C*_30min_	3.62	0.26	-	−0.45	0.84	-	−28.87	0.50	2.46	10.56
	H*_30min_	8.70	0.20	−15.34	−2.94	16.35	0.08	−61.99	0.89	3.41	59.18
LTD	pH_30min_	0.93	0.01	-	−0.01	-	-	0.40	0.92	0.05	180.19
	L*_30min_	−6.10	-	-	-	-	0.43	76.55	0.51	3.66	23.33
	a*_30min_	4.13	-	0.48	-	-	-	−15.75	0.47	1.60	19.98
	b*_30min_	-	-	−2.76	-	2.52	−0.02	7.69	0.45	2.55	12.03
	C*_30min_	4.72	−0.10	−0.07	2.22	−0.21	−0.58	−1.68	0.48	1.62	6.37
	H*_30min_	-	-	−16.59	−7.15	17.05	1.17	13.51	0.43	9.01	8.01
Overall	pH_30min_	1.01	−0.01	0.01	-	-	-	0.13	0.94	0.09	1345.10
	L*_30min_	−2.04	0.91	−0.28	−2.78	1.49	0.79	−5.72	0.81	3.02	179.89
	a*_30min_	−0.33	-	0.48	0.27	-	−0.12	10.71	0.54	1.44	77.40
	b*_30min_	−1.19	0.08	−0.97	0.21	0.97	0.01	6.47	0.54	1.72	51.49
	C*_30min_	−0.64	0.05	-	-	0.56	−0.03	9.08	0.43	1.65	49.33
	H*_30min_	−3.49	0.22	−4.26	−1.45	4.16	0.59	23.89	0.61	5.15	68.67

LD—*Longissimus* *dorsi*, LL—*Longissimus lumborum*, IL—*Iliacus*, SEM—*Semimembranosus*, CT—*Cutaneous trunci*, LTD—*Latissimus dorsi*, L*— white, a*—red, b*—yellow, C*—scale values CIE, H*—scale values CIE, R^2^—determination coefficient, F—statistical test.

## Data Availability

The data presented in this study are available on request from the corresponding author.

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
