# Peer review of "Bloom Time Effect Depends on Muscle Type and May Determine the Results of pH and Color Instrumental Evaluation"

_animals, 2021, doi:10.3390/ani11051282_

Round 1
Reviewer 1 Report
The manuscript is clearly written and well organized. However, there are some minor issues, which need to be improved.
English style and grammar should be corrected as there are some mistakes.
Line 74 if it was divided into two parts it should be: longissimus dorsi lumborum and longissimus dorsi thoracis
Line 83-84 was it 270 carcasses?
I think it should be described more accurately: when was the blooming conducted: was it right after the dissection? and how: was the muscle cut and allowed to bloom? How was it cut? Along or perpendicular to the muscle fibers? etc. this is very important to get this information to be able to repeat the experiment.
Line 89 When was the pH measured?
Line 111 terror? ?
Line 114 RFN should be explained
Line 115-117 What was the other way of confirming the differences than statistically?
Figure 1. The pH values should be indicated in the Figure. Also, there are no letters or asterixes indicated? If there was none, it should be indicated.
Table 1. There are some letters (statistical significances) missing – does it mean there was no difference? It should be explained.
There is no information if the values are means =/- standard deviations or standard errors? The same in Figure 1.
Line 145 should it be except?
Author Response
The answer in the file

Reviewer 2 Report
The manuscript presents some aspects of interest. Certainly, biochemical changes of muscles and the effect of blooming is a matter of interest and there is a work to be done. However, there are some aspects the I believe have not been addressed adequately.
Main concern.
Statistical analysis. Please, define the experimental unit in this experiment. Is it the muscle? Statistical analysis of sampling the same experimental unit over the time should be studied by a repeated measurement test procedure in which the effect of treatment and time (and its interaction) is obtained. Please, consider it. I do believe, this kind of analysis would clarify some confusing information (with all these lettering labels in Table 1) and may allow to obtain clearer conclusions.
pH. Please describe in Material and Method how pH was measured. Was it measures in the surface? Was it measured internally by puncture? If so, please indicate how deep was it introduced. Figure 1 does not show units. Again in this particular piece of information (which I believe is relevant) I believe the repeated measurement test would allow obtaining a much better overview, as I would expect an muscle effect, a time effect and an interaction). These results regarding pH change after cutting are nor adequately discussed. Please provide similar results from other authors, compare them and discuss it.
Sampling. Please, explain how muscle sampling was done. Were samples obtained directly form the carcass, or were obtained at cutting? Was it done on darkness? Did you get a bigger piece of the muscle at cutting, and then took it to the lab to get a new slice where you carried out all further analysis? This is a critical point and I do believe should be described in full detail.
Please, also explain the potential interest of this piece of research. What is the interest of studying changes along blooming? Please, also try to focus discussion on this possible interest. Otherwise, the interest of the manuscript is limited to some relationship among variables with limited interest. I do believe, the interest of your data is higher than the information that can be obtained from the discussion.
Minor concern.
The section ‘4. Discussion’ is hidden in a footnote and it took ma time to understand the structure of the manuscript. Please, correct it.
Define blooming and explain its possible interest for meat science from initial paragraph in the introduction.
I am not native in English but I can see a number of mistakes (e.g the need to accurately avaluation [I would say evaluate], constantly [constant] changes; bloom time were [is] showed; muscles were [are] presented; division for muscles were (are] presented in Figure 3; in relation to the changing [change] of pH, and a number of other). Please, correct it.
Figure 3 is difficult to understand. I suggest removing it and present regression analysis instead.
Please re-write conclusion and focus it on your particular results. The statements you include here ‘The information may be particularly important for mat plants, …’ is adequate for introduction section. In the conclusion, I would expect a more precise statement on the potential interest of your results. (I believe you can provide a nice conclusion, as I believe you results are nice), so I encourage you to work on it.
Author Response
The answer in the file

Reviewer 3 Report
Overall, it is good paper in the existing conditions. However, i have below comments for authors to consider:
- In Materials and Methods: can you add some information about where animals were reared or how transported and also how much time they spend in lairage? to know any possible effect on quality before slaughteirng?
- in sentence: The cooling of car- 80 casses was performed first by passing them for 2 h at -15 to -8°C and followed by storage 81
at 0–2°C (8–12 h), is it according to the standard protocols of slaughterhouse or you have modified some condition? if you have modifed then why? - All figures are in black and white; it is very hard to differetiate different columns, can you make colors figiures? it will be then easier for readers to get the message quickly. I may suggest that you must have to change them into color figures or make them easy for the readers.
Author Response
The answer in the file
